# Issues related to pregnancy, pregnancy prevention and abortion in the context of the COVID-19 pandemic: a WHO qualitative study protocol

Jose Guilherme Cecatti ![ORCID],[1] Luis Bahamondes ![ORCID],[1] Moazzam Ali ![ORCID],[2] Deda Ogum Alangea,[3] Vanessa Brizuela ![ORCID],[4] Eunice Nahyuha Chomi,[5] Seni Kouanda,[5] Rozina Karmaliani,[6] Laila Ladak ![ORCID],[6] Pisake Lumbiganon,[7] Modey Emefa,[8] Sothornwit Jen,[9] Hamsadvani Kuganantham,[4] Caron Kim ![ORCID],[2] WHO HRP Social Science Research Team

For numbered affiliations see end of article.

**Correspondence to**
Dr Caron Kim; kimca@who.int

## ABSTRACT

**Introduction** WHO has generated standardised clinical and epidemiological research protocols to address key public health questions for SARS-CoV-2 (COVID-19) pandemic. We present a standardised protocol with the aim to fill a gap in understanding the needs, attitudes and practices related to sexual and reproductive health in the context of COVID-19 pandemic, focusing on pregnancy, pregnancy prevention and abortion.

**Methods and analysis plan** This protocol is a prospective qualitative research, using semi-structured interviews with at least 15 pregnant women at different gestational ages and after delivery, 6 months apart from the first interview. At least 10 partners, 10 non-pregnant women and 5 healthcare professionals will be interviewed once during the course of the research. Higher number of subjects may be needed if a saturation is not achieved with these numbers. Data collection will be performed in a standardised way by skilled trained interviewers using written notes or audio-record of the interview. The data will be explored using the thematic content analysis and the researchers will look for broad patterns, generalisations or theories from these categories.

**Ethics and dissemination** The current protocol was first technically assessed and approved by the WHO scientific committee and then approved by its ethics review committee as a guidance document. It is expected that each country/setting implementing such a generic protocol adapted to their conditions also obtain local ethical approval. Comments for the user's consideration are provided the document, as the user may need to modify methods slightly because of the local context in which this study will be carried out.

## STRENGTHS AND LIMITATIONS OF THIS STUDY

⇒ This is part of a series of WHO generic protocols with the purpose of facilitating and standardising the assessment of sexual and reproductive health issues during and after the COVID-19 pandemic in low-income and middle-income countries.

⇒ With a generic protocol, adaptations of it are essential and encouraged, depending on the characteristics and resources from the countries/settings, where they are planned to be implemented or other places deciding to do the same.

⇒ One study limitation would be the difficulties and risks of different contexts of participating centres' places regarding the COVID-19 pandemic at the moment of study implementation, which could introduce biases into subjects' perception.

## INTRODUCTION

According to the WHO, as of 18 September 2022, around 609 million cases and 6.5 million deaths of SARS-CoV-2 (COVID-19) cases have been confirmed globally. COVID-19 was declared as a public health emergency of international concern at the end of January and characterised as a pandemic by WHO on 11 March 2020.[1]

China was the first country heavily affected by the pandemic since December 2019,[2] soon followed by several others in Asia, and then successively in Europe, America and Africa. To restrain the spread of COVID-19, domestic mobility was strictly regulated and in some affected areas transportation was restricted, with some differences in strategies adopted by different countries in different timeframes. In addition, physical distancing measures and isolation were firmly recommended in many countries and territories in different degrees, including closure of some basic services such as in-person classes at schools and universities, business activities, commerce, industry and even some healthcare services. As a result, there is some information indicating a disrupted healthcare service provision, including services in sexual and reproductive health (SRH).[3 4] Despite the enormous medical and research

resources directed toward the COVID-19 response, there are several gaps in the knowledge with regards to how the virus affects the body, even after recovery, and how the disease and the pandemic has affected people's lives.

Given SARS-CoV-2 novelty as virus of human transmission, there is limited evidence available to identify the impact of COVID-19 on SRH. At the start of the pandemic, there were questions whether or not the transmission of the virus or disease occurs from the pregnant woman to the fetus including at the early stage of fetal development.[5] The evidence has been growing on the risk of mother-to-child transmission, whether during pregnancy, childbirth or breastfeeding.[6–9] In any case, the potential effects of COVID-19 infection during pregnancy on maternal and fetal health would influence women, their partners and healthcare professionals' decisions on reproductive healthcare.

According to the UNFPA, as health systems became overwhelmed with COVID-19 cases, clinical staff may not have had the time or personal protective equipment (PPE) needed to provide contraceptive guidance and provision.[10] Women were and are refraining from visiting healthcare facilities due to movement restrictions or fears about COVID-19 exposure. Supply chain disruptions are limiting the availability of contraceptives in many places. It is anticipated that significant levels of lockdown-related disruption over the months since the pandemic was declared could leave more than 47 million women in low-income and middle-income countries unable to use modern contraceptives, leading to a projected 7 million additional unintended pregnancies.[10]

In places in which medical termination of pregnancy is allowed, it has also been suggested that the demands for these services, including information provision, have increased, which may be related to lack of contraceptive commodities or fear of unknown consequences of infection during pregnancy in the context of pandemic. There is some evidence from lower-resourced settings of women, especially those in an established relationship, having less control over the decision-making process of contraception or abortion.[11 12] Their husbands/partners and the extended family members have more control over their reproductive autonomy. It is important to examine how COVID-19 and the lack of resources during the pandemic influenced the decision-making process and the power dynamics involved in accessing contraception and abortion using information already available or implementing surveys for this.[13 14]

Thus, it is important to assess the behavioural and psychosocial impacts of the COVID-19 pandemic on some aspects of women's SRH, especially among those who are pregnant, who are planning to become pregnant and those who want to avoid pregnancy, together with their partners and healthcare professionals, with a special attention to domestic and gender-based violence (GBV).[15]

The current community-based protocol covers various SRH issues: abortion, contraception, pregnancy, sexually transmitted diseases prevention, domestic violence and GBV, stigma and discrimination. Covering multiple aspects of SRH will provide a better understanding of the status quo of SRH under the COVID-19 pandemic.

**Comment**: this section should be updated with country/setting-specific and updated data on COVID-19 epidemiology and current research findings prior to submission to local/national Institutional Review Boards (IRB) or donor agencies. Specific information regarding the health system management and provision of services should also be included in this section (facility-based childbirths, access to safe and legal abortion services and access to contraception).

## Problem statement, aims and objectives

The global research question is on what is the impact of COVID-19 on the SRH of women (we recognise that cisgender women, transgender men, non-binary, gender-fluid and intersex individuals with a female reproductive system and capable of becoming pregnant may require SRH care. To facilitate readability of this protocol, when referring to all gender diverse people who may require SRH services, we used the word 'women' most often, although we also use the term 'subjects' and 'individual') and their partners. The main aim is to increase knowledge and understanding of the behavioural and psychosocial impacts of the pandemic on women and men's SRH, especially those related to pregnancy, pregnancy prevention and abortion.

Their specific objectives are: (1) to investigate how the existence of the COVID-19 pandemic affects women's knowledge, attitudes, needs and practices on SRH services such as pregnancy care, pregnancy prevention, and abortion; (2) to explore how the pandemic affects men's and healthcare professionals' attitudes, needs and practices related to pregnancy, pregnancy prevention, and abortion; and (3) to understand the knowledge and the perceptions of the study population on the COVID-19 impact and how this relates to their SRH-related attitudes and behaviours.

**Comment:** the SRH topics that the local study team intends to explore should be inserted in the objectives and may vary widely depending on the characteristics of the setting. Additional secondary objectives can be included, depending on the local context, in addition to the willingness of researchers and availability or resources to expand the current proposal.

## METHODS
### Study design
This study will use a phenomenological qualitative design with semi-structured interviews to collect data in either one point in time (for the subjects' partners and health workers) or two points in time (for women), 6 months apart. It is planned to be conducted in selected urban, semi-urban and possibly rural regions and will recruit a purposive sample of pregnant and non-pregnant subjects

to enable researchers to explore women's knowledge, attitudes and practices in different temporal and situational contexts under COVID-19, including diversity in pregnancy status, SRH needs and socioeconomic status.[16] In addition, their partners and local healthcare professionals will also be recruited to participate.

## Study settings

**Comment**: there should be a brief description of the location—country, state, province and city—where the study is planned to be implemented, with some arguments justifying why this choice was made. Inform where and how pregnant women have access to antenatal care, where and how they deliver and access postnatal care, and if, where and how they get access to abortion and family planning services. State the number of health facilities chosen and their locations.

This protocol was developed to be implemented mainly in low-income and middle-income countries with institutions that have traditionally been partners in research with the WHO/HRP. However, any country or research centre can use the same protocol and they will be very welcomed. To capture variation in responses, health facilities in both rural and/or semi-urban/urban settings will be selected as focal points, including primary health unit or hospital, where pregnant women are usually required to seek antenatal services. This enables the researchers' easy access to individuals attending the facility for interviews. The focal facilities will be selected purposively, using such criteria as available resources, COVID-19 pandemic status, availability of SRH services and the willingness of the healthcare providers to participate. The final selection of study sites will be confirmed jointly by the research team and local authorities, depending on the national procedures and regulations.

**Comment:** the implementation protocol should clearly define the catchment area of included health facilities and describe the demographics of the study sample. The strategy for getting help from healthcare providers at health facilities or other ways of identifying eligible subjects will rely on local characteristic of the health system and should be detailed described.

## Study participants and sampling

The healthcare providers of the facilities selected as focal points will be asked to help the research team identify potential participants attending the facility. Pregnant individuals will be recruited at the local healthcare clinics when they access antenatal and/or SRH care services. Other participants, that is, non-pregnant individuals, partners of pregnant and non-pregnant individuals and healthcare professionals, will be recruited from these same selected facilities during clinical visits. To explore contraceptive choices, abortion care and pregnancy care, different women will need to be selected accordingly. Therefore, the location sampling method (venue based) will be used to recruit the participants,[16] where participants can be selected from places they attend, which can

be identified in advance. This location sampling process is easily implemented in healthcare facilities, where a sufficient number of available participants for the study are supposed to be present at an acceptable period of time.

Researchers will approach all potential participants following standard inclusion criteria. For pregnant women, the inclusion criteria include: (1) age 18–45 years old, (2) living within the catchment area of the healthcare facility, (3) who were pregnant (ideally, in early pregnancy, in the first trimester) during the COVID-19 outbreak in that specific location (level 3–4 on the WHO Situational Level Assessment) and (4) either who have tested positive or negative for SARS-CoV-2 but currently not with an active infection (ie, current PCR+ cases or within 2 weeks from the beginning of symptoms will not be recruited in the study while in this condition). For non-pregnant women, the inclusion criteria include: (1) aged 18–45 years old, (2) living within the catchment area of the facility, (3) who are not pregnant during and immediately after the COVID-19 outbreak in that location at the moment of enrolment and (4) who are attending the health facility with the purpose of getting any service on SRH women who do not meet the above criteria or who do not wish to take part will be excluded. Meanwhile, partners/family members of women in both groups attending the healthcare facility as companions and healthcare professionals from these facilities will be invited to participate in the interviews. Participation will be voluntary and only individuals who consent to participate after receiving information will be enrolled. Informed consent will be administered to the study participants in their local language highlighting the details about the study objectives and processes. Signed informed consents forms will be mandatory for face-to-face interviews. Otherwise, if interview is performed by phone or other electronic means, a verbal consent will be obtained; this will be necessary if the participant is not literate or has visual impairment. To account for the possible loss to follow-up of pregnant women, we will recruit 20% more participants than the intended sample size during the first interview.

**Comment**: the study will use gatekeepers who will be health service providers not participating in the study or community members who are known and trusted by the clients. Their main role will be to limited to identification and introduction of potential participants to the study team.

## Sample size calculation

This study will recruit a sample of a minimum of 15 pregnant women per facility. Specifically, the suggested sample size is an approximation to the lower bound (the minimum number or participants required). However, the upper bound is determined by sampling until saturation, which means they should be selected until it is found that similar answers are given to similar questions with no new themes emerging. The decision on saturation is expected to be taken according the experienced

and trained interviewers/research assistants under the supervision of the local principal investigator (PI) and other possible skilled participating researchers. Sample sizes can vary largely depending on the context that could interfere with the saturation of gathered information. In addition, at least 10 non-pregnant women, 10 partners (any gender, for both pregnant and non-pregnant women) and 5 healthcare professionals will also be recruited in each country. The minimum sample sizes were not determined by any specific criterion, but using a priori estimation. This was empirically decided as the minimum number of subjects in each category to provide meaningful information to be gathered with that from other countries/settings. A purposive sampling strategy aimed at saturation on the central study themes is appropriate for a qualitative study.

**Comment**: the sample size estimation should roughly follow the above-mentioned process, but adaptations are possible depending on the local characteristics and timeframe relative to the COVID-19 pandemic.

Adjustments should also be made if testing for infection is unavailable. Recruitment can be undertaken with assessment of infection status based on history (ie, lack of symptoms or no report of being in close contact with someone with COVID-19)

### Patient and public involvement
There was no patient or public involvement in the design of this protocol.

### Data collection methods
This study includes a prospective series of two semi-structured interviews to be conducted with pregnant women and one semi-structured interview to be conducted with non-pregnant women, partners of both pregnant and non-pregnant women, and healthcare professionals in selected study areas. The reason for conducting two interviews with pregnant women is to explore the short-term and long-term impact and changes of antenatal care, abortion and potential experiences with childbirth. Each interview will last approximately 60–90 min. The script of the interviews will follow a topic guide and the researchers will promote a safe, convenient and comfortable environment to enable a comprehensive rapport. The topic guide will be pretested to ensure question validity and assess interview length. During the interview, the research team will ask open-ended questions to participants and record field notes.

The process of participant recruitment aims to provide comprehensive information about the study as well as to build trust and rapport with the participants, as an essential step for them to make a fully informed and voluntary decision. This process will be documented with a signed written or oral recorded informed consent. The contact information of the enrolled pregnant women will be asked (eg, phone number or other forms of preferred contact), as well as that of other contacts (eg, relatives) during the first interview to improve chances for follow-up for the

second interview. Follow-up and the second interview will be performed in person or through phone/teleconference based on the participant's preference and need.

The interviews will ask about fertility desires, pregnancy, pregnancy prevention, abortion and their related needs for accurate information and reproductive services. The first interview will capture how the above-mentioned factors, especially the needs for reproductive services, were affected by COVID-19. The second interview will assess how the above-mentioned factors changed during the outbreak. The semi-structured interview allows for exploration of other SRH issues if these come up. The interviews will document pregnant women's knowledge, concerns, needs, attitudes and practices throughout the duration of their pregnancy on SRH, abortion, contraception, GBV, etc.

A single semi-structured interview will be conducted with the other participants at one point in time. Interviews with non-pregnant women will document their knowledge, attitudes and practices toward SRH under the context of COVID-19. Interviews with partners will explore their knowledge, attitudes and concerns for their own and partner's SRH and well-being, including abortion. Depending on the context, local researchers may wish to interview extended family members (such as in-laws) in addition or instead of partners, as indicated by local customs. Interviews with healthcare professionals will focus on access to and availability of contraception and safe abortion; health-systems versus women's roles and responsibilities for prevention of pregnancy in the context of COVID-19; health system capacity to provide high-quality care for women; attitudes about pregnancy, contraception and abortion in the context of COVID-19; and perceived psychosocial effects on women, their families and local communities.

The study instruments were adapted from a core master guide built by WHO for qualitative research on women's reproductive and psychosocial health. In particular, questions were adapted to the context of COVID-19, but they should also include items relevant to the local settings. The instrument will be translated into the language of the location and pretested to ensure its validity and reliability with adjustments if needed. The interview core and detailed guides are present in online supplemental files S1 and S2.

Interviewers may have to confront difficult situations among infected participant cohort—like death or serious complications from COVID-19. The pandemic has changed the way the people live, exposed most of the population to unsafe and difficult situations which has affected many people personally (either because they themselves were sick or their close ones, or because they might have lost close ones to the disease). Similar sensitive situations could also appear regarding GBV and child abuse for instance. Ideally the study team members should have access to a mental healthcare provider for counselling and support to researchers and interviewees experiencing difficult situations. The country coordinating

team should be able to provide the participants in the study with appropriate recommendations on how and where to access specific services if problems arise during the interviews.

Several changes occurred in the dynamics of the pandemic since its beginning. The protocol will probably be implemented in several different countries experiencing different stages of the pandemic. This is a potential factor to enrich the content of the findings from different settings. In addition, current experience on clinical and research activities show that such an interview is possible with PPE or using interviewers who have already had COVID-19 or who were already vaccinated.

**Comment:** depending on the local experience and availability of resources, the interviews should ideally be recorded and then appropriate data management with records, transcription, use of specific software and related topics should be adapted in the current protocol. If audio-recording is not an option, then data collection and recording of the interview will need to be adapted using hand notes. Additional themes can be added to the interview as appropriate for each target group and questions revised accordingly, across the whole study. Interview tools and the informed consent form would also need to be adjusted in settings where abortion is legally restrictive, so that the questions are posed in a sensitive and non-judgemental manner. Given the inclusion of sensitive topics, it is essential that the study team implements all measures to protect the privacy and confidentiality of participants. The guidance document (online supplemental files) should provide potential options to countries on which questions to eliminate or modify to fit the local context. This should also be considered in the interview guide. Similar careful attention should be given to questions related to violence.

### Data analyses

We adopt concepts from the ecological theory[17] to approach this research question. The ecological theory suggests that individual characteristics as well as the environment act interactively on one's personal development. In this context, individual characteristics refer to one's social-demographic characteristics, and one's knowledge and attitude related to SRH. The environment refers to a broad range of social and structural factors, including societal value, and access to SRH services. COVID-19 is the overarching environmental factor of the proposed study. This model serves as the framework for studying the knowledge, needs, attitudes and practices related to contraception, abortion and healthcare before, during and after pregnancy in the context of the COVID-19 pandemic. This model will also guide study design, data collection, analysis and interpretation.

An inductive analytical technique will be used to analyse the data collected. Detailed examination of documents from interviews, either from written forms or transcription from recordings, will be conducted in order to generate initial categories and, above all, to suggest relationships

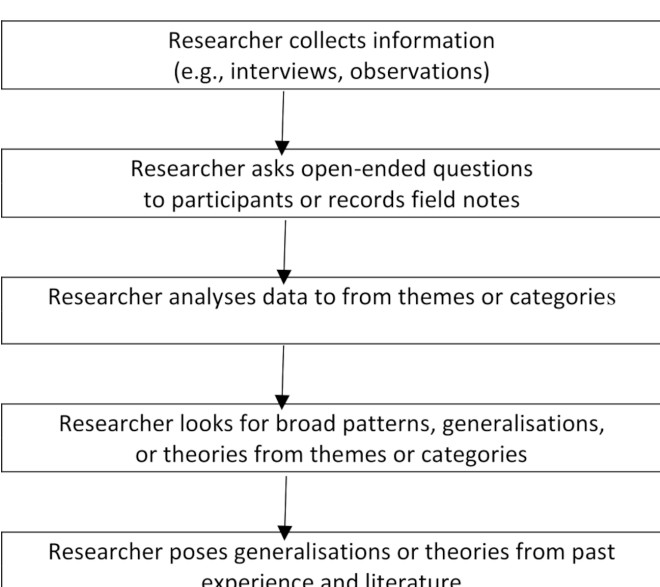

**Figure 1** The inductive logic of research in a qualitative study.

between categories. Two researchers will do an initial screening of the interview transcripts and develop codes. During the process, the researchers will compile a list of themes and write short notes about important issues and patterns that are emerging. From these new themes, the researchers establish inductive codes. The researchers will then go through the transcripts again, and recode them according to the new codes and themes. The researchers will also compile all the above-mentioned codes into a codebook. Two researchers will analyse the data concurrently either manually or with the help of a qualitative analysis software (such as NVivo). The coding method and progress will be systematically discussed between the researchers and a third team member to ensure intercoder reliability.

There are several optional techniques available for identifying core categories and their integration, such as data usage diagrams and written memos. Figure 1 provides a diagram with the elements of the procedures and process of applying the conceptual framework to this qualitative study.

There are different parties involved in this study as individuals living in COVID-19 pandemic areas during pregnancy, the SRH needs of non-pregnant women and their partners, including safe sex, contraception and abortion, as well as SRH healthcare providers. The participating partners in the study will be asked about their own SRH needs as well as those of their pregnant/not-pregnant partner.

Although the analysis is planned to provide a general full picture of SRH for women and men across all the participating countries/sites, the conceptual analysis will not be standardised across diverse contexts/countries due to their specificities and probable local adaptations for implementing the research protocol.

## Data management and data access

Audio recordings and/or notes from the interviews will be transcribed verbatim, cross-checked as soon as feasible following the interview. Only the research team will have access to the data during the study. Single interviews will be de-identified at the time of transcription. Prospective interviews will be de-identified, transcribed, cross-checked and summarised in order to inform subsequent interviews in the two-interview series. A participant ID will be assigned to each participant to ensure that data from the same participant is linked across two interview series. Data and transcriptions will be kept in encrypted hard drives, accessible only to the research team. All audio recordings will be kept in a secure place for the period of time as required for local country regulations and will then be destroyed. A subsample of the interview transcriptions will be translated into English (for those performed in a different language) to enable monitoring and inputs from WHO staff and consultants. All identifiable information, including telephone number and electronic accounts, which are used for contacting subjects during the interview, will be stored in a separate encrypted hard drive.

## Ethics and dissemination

Respondents will provide a signed informed consent, also allowing eventually for a telephone interview (online supplemental files S3). The information collected will be de-identified. The identifiable information will be stored in a separate encrypted hard drive. The data will be kept in a safe storage way for the time as required by local national rules and laws on this topic. If additional analyses are planned, an appropriate permission should be obtained from the local ethics committees. If a participant decides to withdraw, the participant will be discontinued and data already collected will be used. However, if the participant withdraws the consent, the respective data will be excluded and will not be used in the analysis.

Although the study is not planning to interview subjects currently with an active PCR+, all safety measures should be taken to minimise transmission risks. Masks will always be a requirement for both researchers and study participants and possibly other PPE and physical distancing as locally required. In cases of suspected infection in either the researcher or the interviewee, they should be immediately referred to the health facility to be screened and tested, and the interview should be interrupted or postponed. If GBV is being assessed and women are identified in such a condition, social and legal advice should also be provided following the recommendations of local IRBs. In sites where abortions are illegal, discussions of access to abortion or postabortion care may include risks to participants in terms of potential personal or legal repercussions.

There will be no financial incentives provided to participate in this study. However, the participants and their companions could receive compensation to cover the incidental costs of their participation, depending on the local regulations and availability of resources. This study protocol has received scientific approval through HRP's research project review panel (RP2) and ethics review committee and, in accordance to local regulations, should be reviewed by a local ethics committee.

The results will be used to build an understanding of knowledge, practices, needs and attitudes related to fertility desires, pregnancy, pregnancy prevention and abortion under the context of COVID-19. They can help to improve health policies and SRH services. We aim to disseminate results with service providers, policy-makers and communities so as to provide them with evidence of what women and their partners think about COVID-19, and suggestions for them to better provide services during future emergencies. Study results will be disseminated through online platforms, as well as publications on international peer-reviewed journals. This study aims to collaborate with potential research capacity strengthening for eligible partners in the country, including any National Research Council, national federation of professional specialties, etc. By participating in the data collection, analysis and interpretation, a group of young national scholars will be trained in qualitative research skills in the field of SRH.

**Comment**: for local adaptations, ensure to note additional precautions and preparations required by study teams in countries where abortion is illegal and/or restricted by their laws and policies. This should include heightened sensitivity to protection of participants' confidentiality (women, men and healthcare workers) and great care in de-identifying healthcare workers' comments regarding referral for termination in countries where illegal and/or subject to religious sanctions. The same applies for GBV.

**Author affiliations**
[1]Department of Obstetrics and Gynecology, University of Campinas, Campinas, Brazil
[2]UNDP/UNFPA/UNICEF/WHO/World Bank Special Programme of Research, Development and Research Training in Human Reproduction (HRP) Department of Sexual and Reproductive Health and Research, World Health Organization, Geneva, Switzerland
[3]Department of Population, Family and Reproductive Health, College of Health Sciences, University of Ghana, Accra, Ghana
[4]UNDP/UNFPA/UNICEF/WHO/World Bank Special Programme of Research, Development and Research Training in Human Reproduction (HRP) Department of Sexual and Reproductive Health and Research, WHO, Geneve, Switzerland
[5]Institut de Recherche en Sciences de la Sante, Ouagadougou, Burkina Faso
[6]Aga Khan University School of Nursing & Midwifery, Karachi, Pakistan
[7]Department of Obstetrics & Gynecology, Khon Kaen University, Khon Kaen, Thailand
[8]Dept. of Population, Family & Reproductive Health, College of Health Sciences, University of Ghana, Accra, Ghana
[9]Department of Obstetrics and Gynaecology, Khon Kaen University, Khon Kaen, Thailand

**Acknowledgements** The authors thank the UNDP/UNFPA/UNICEF/WHO/World Bank Special Programme of Research, Development and Research Training in Human Reproduction (HRP), Department of Sexual and Reproductive Health and Research and the World Health Emergency programme from the WHO for leading and supporting this initiative of developing a generic protocol for such a qualitative study plus the publication of this manuscript. The authors also thank the overseas

members of the hubs from the HRP Alliance, who supported the initiative and gave important contributions to the protocol.

**Collaborators** WHO HRP Social Science Research Team: Brazil (Luis Bahamondes and Jose Guilherme Cecatti); Burkina Faso (Eunice Chomi and Seni Kouanda), China (Kun Tang, Yueping Guo, Yifang, Hanxiyue Zhang, Yifan Zhu, Ge Yang and Chunxiao Peng), Ghana (Deda Ogum Alangea, Kwasi Tropsey and Emefa Judith Modey), Pakistan (Rozina Karmaliani and Laila Ladak), Thailand (Pisake Lumbiganon and Jen Sothornwit) and WHO Secretariat (Moazzam Ali, Caron Kim, Hamsadvani Kuganantham, Vanessa Brizuela, Anna Thorson, Joy Jerop Chebet, Hugo Gamerro Abrego, Soe Soe Thwin and Armando Seuc).

**Contributors** CRK and MA conceptualised and designed the protocol. JGC and LB performed the first round of adaptation of the protocol to this manuscript under the supervision of CRK and MA. All other authors, DOA, VB, EC, SK, RK, LL, PL, EJM, JS and HK, participated with important suggestions for the proposed adaptation to local settings, read and approved the final written manuscript.

**Funding** This research has been supported by the German Federal Ministry of Health (BMG Germany) COVID-19 Research and development funding to WHO (award: 70918). It also has been supported by the UNDP/UNFPA/UNICEF/WHO/ World Bank Special Programme of Research, Development and Research Training in Human Reproduction (HRP), a co-sponsored programme executed by the WHO. They are also funding the adaptation and implementation of the protocol in the HRP collaborating centres in Brazil, Burkina Faso, China, Ghana, Pakistan and Thailand. The views of the funding body have not influenced the content of this manuscript. This article represents the views of the named authors only and does not represent the views of their affiliated institutions.

**Disclaimer** The authors alone are responsible for the views expressed in this article and they do not necessarily represent the views, decisions or policies of the institutions with which they are affiliated.

**Patient and public involvement** Patients and/or the public were not involved in the design, or conduct, or reporting, or dissemination plans of this research.

**Provenance and peer review** Not commissioned; externally peer reviewed.

**ORCID iDs**
Jose Guilherme Cecatti http://orcid.org/0000-0003-1285-8445
Luis Bahamondes http://orcid.org/0000-0002-7356-8428
Moazzam Ali http://orcid.org/0000-0001-6949-8976
Vanessa Brizuela http://orcid.org/0000-0002-4860-0828
Laila Ladak http://orcid.org/0000-0002-7015-0413
Caron Kim http://orcid.org/0000-0002-3574-4160

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
