## [Reviewer comments · BMJ Open]

ARTICLE DETAILS

TITLE (PROVISIONAL)	Issues related to Pregnancy, Pregnancy Prevention and Abortion in the Context of the COVID-19 pandemic: a WHO qualitative study protocol
AUTHORS	Cecatti, Jose; Bahamondes, Luis; Ali, Moazzam; Alangea, Deda Ogum; Brizuela, Vanessa; Nahyuha Chomi, Eunice; Kouanda, Seni; Karmaliani, Rozina; Ladak, Laila; Lumbiganon, Pisake; Emefa, Modey; Jen, Sothornwit; Kuganantham, Hamsadvani; Kim, Caron; Collaborator Group, WHO HRP Social Science Research Team

VERSION 1 – REVIEW

REVIEWER	Fathnezhad-Kazemi, Azita Islamic Azad University
REVIEW RETURNED	04-Nov-2021

GENERAL COMMENTS	There are some confusing issues in your manuscript, first of all, according to the abstract, the aim of the study is “fill a gap in understanding the needs, attitudes, and practices related to sexual and reproductive health (SRH)” so it is different from the title and the purpose mentioned in the problem statement section “women’s knowledge, attitudes, needs and practices on SRH services” and I couldn’t understand how you will investigate knowledge about COVID-19 with qualitative study design. Second, in my opinion, all women at different stages of their life have different needs and attitudes about everything so how you will integrate all data one more thing inclusion and exclusion criteria Not properly specified especially in the non-pregnant women. How will you control the validity and reliability of data? Achieve the saturation of data is important in the qualitative study design so How do you estimate the sample size and calculate the drop rate? It seems that there are a lot of biases in the study, there are different parts and a lot of questions.
---

REVIEWER	Hedriana, Herman University of California Davis, Department of Obstetrics and Gynecology
REVIEW RETURNED	08-Nov-2021

GENERAL COMMENTS	Cecatti et al presented a protocol to qualitatively assess various issues to pregnancy, pregnancy prevention and abortion during the COVID-19 pandemic on behalf of a WHO special program. The impetus is to fill a gap in understanding the needs, attitudes and practices related to sexual and reproductive health (SRH)
---

	during the COVID pandemic. The focus is on healthcare services disruption in SRH. Overall, the protocol is written well, and I look forward to the study results. Here are my comments:  1. Introduction is too verbose. Need to shorten and state an argument for the project without too much historical detail. 2. Page 6, line 15, what is SRHR? The prior abbreviated reference was for SRH. 3. Please restate specific objective 3. As it appears, the syntax is confusing and appears to be out of context. Do the authors want to understand the subjects' knowledge and their biases towards COVID? 4. From the study design, it appears the study is structured for heterosexual partners. If not, line 46, page 6, needs clarification regarding partners. Also, the rationale of 1-point vs. 2-points in women. 5. Line 55, page 6 - the local health professional and "partners" (gender preference consideration?) recruited to participate how? 6. Please clarify study setting - what is preferred or the criteria to be chosen as a setting 7. Line 58, page 7 - "required" does not sound appropriate. Suggest "available" 8. The authors will need to expound on the sample size calculation especially how the lower bound was determined at 15 for pregnancy, 10 for non-pregnant, 10 for partners (with and without pregnancy???) and 5 healthcare professionals (at what level of healthcare?) 9. Page 10, line 20 - What is GBV? This appeared without definition. again, in page 11, line 5. 10. Page 15, line 50/51 - salute the authors to think beyond heterosexual partnership. Would suggest encompassing a nongender approach knowing that only biological females can become pregnant
--	--

VERSION 1 – AUTHOR RESPONSE

Reviewer: 1

Dr. Azita Fathnezhad-Kazemi, Islamic Azad University

Comments to the Author:

Dear author

There are some confusing issues in your manuscript, first of all, according to the abstract, the aim of the study is “fill a gap in understanding the needs, attitudes, and practices related to sexual and reproductive health (SRH)” so it is different from the title and the purpose mentioned in the problem statement section “women’s knowledge, attitudes, needs and practices on SRH services” and I couldn’t understand how you will investigate knowledge about COVID-19 with qualitative study design.

We agree that they are different, but we believe they are related. All these objectives are planned to be related to pregnancy, pregnancy prevention and abortion. We have ensured to clearly state this in the abstract and also in the problem statement section. It is further highlighted in the second paragraph of the same section. We are not going to investigate the knowledge specifically about COVID-19, but what women and men feel regarding its impact in their SRH. This misunderstanding probably was due to the phrasing of objective 3 which has now been reformulated.

Second, in my opinion, all women at different stages of their life have different needs and attitudes about everything so how you will integrate all data. One more thing, the inclusion and exclusion criteria are not properly specified, especially in the non-pregnant women.

Thank you. We agree with you and exactly because of this we believe there is a need to investigate these “different stages of women’s life”. Although the numbers are relatively low, please remember that it will be implemented in several different countries/settings and therefore data from similar women can be joined to better characterize the SRH needs at different stages of life.

The inclusion/exclusion criteria for non-pregnant women are on the second paragraph of the section “Study participants and sampling”. We agree that perhaps they are not clear enough and have modified the text. These women in reproductive age who are not pregnant and living in the neighbourhood of the health facility should also be attending the health facility looking for services or advice on SRH. This is now clearly stated in sub-item 4.

How will you control the validity and reliability of data?

As a qualitative study, no formal assessment of validity and reliability of data are planned. However, the pretest of the translated original instrument into local languages is necessary to assure their validity and reliability as a common qualitative procedure. This is explained in the fifth paragraph of the section “Data collection methods”.

Achieve the saturation of data is important in the qualitative study design so How do you estimate the sample size and calculate the drop rate?

We do agree that saturation is important in a qualitative study design. Now we included the information “The decision on saturation is expected to be taken according to the experienced and trained interviewers/research assistants under the supervision of the local PI and other possible skilled participating researchers” in the paragraph of the topic “Sample size calculation”. In addition, the same paragraph explains that sample size was estimated a priori, without any specific criteria, as commonly done for qualitative studies. Apart from the pregnant women who will be interviewed twice (the second time six months after the first), there will be no drop rates because individuals will be interviewed just once. For pregnant women we estimated a high rate of loss to follow up of 50%, considering the local difficulties for postpartum women participating in studies.

It seems that there are a lot of biases in the study, there are different parts and a lot of questions.

Unfortunately, we cannot address specifically these points that were not identified

Reviewer: 2

Prof. Herman Hedriana, University of California Davis, UC Davis Children's Hospital

Comments to the Author:

Cecatti et al presented a protocol to qualitatively assess various issues to pregnancy, pregnancy prevention and abortion during the COVID-19 pandemic on behalf of a WHO special program. The impetus is to fill a gap in understanding the needs, attitudes and practices related to sexual and reproductive health (SRH) during the COVID pandemic. The focus is on healthcare services disruption in SRH. Overall, the protocol is written well, and I look forward to the study results. Here are my comments:

1. Introduction is too verbose. Need to shorten and state an argument for the project without too much historical detail.

Thank you for this valid point which the research group had discussed extensively. The final argument was that this was not a simple research protocol from a single-centre, but a WHO protocol ready to be implemented in several countries and also providing a context for stimulating others to develop similar protocols in their settings.

2. Page 6, line 15, what is SRHR? The prior abbreviated reference was for SRH.

Thank you for noting. This was a mistake. The correct is SRH, now changed.

3. Please restate specific objective 3. As it appears, the syntax is confusing and appears to be out of context. Do the authors want to understand the subjects' knowledge and their biases towards COVID? **Thank you for pointing this out. You are right. We do not want to assess the subjects' knowledges about COVID but its effects. This is now reworded.**

4. From the study design, it appears the study is structured for heterosexual partners. If not, line 46, page 6, needs clarification regarding partners. Also, the rationale of 1-point vs. 2-points in women.

This is one of the reasons that the term “partner” was used to be gender-neutral. The study is structured for both male partners and for female partners. This is now clearer in the manuscript.

5. Line 55, page 6 - the local health professional and "partners" (gender preference consideration?) recruited to participate how?

This has been clarified in the manuscript.

6. Please clarify the study setting - what is preferred or the criteria to be chosen as a setting.

The following phrase was included in the beginning of the paragraph: This protocol was developed to be implemented mainly in some low and middle-income countries with institutions that have traditionally been partners in research with the WHO/HRP. However, any country or research centre can use the same protocol and they will be very welcome.

7. Line 58, page 7 - "required" does not sound appropriate. Suggest "available"

Agreed. Changed accordingly.

8. The authors will need to expound on the sample size calculation especially how the lower bound was determined at 15 for pregnancy, 10 for non-pregnant, 10 for partners (with and without pregnancy???) and 5 healthcare professionals (at what level of healthcare?)

This was empirically decided as the minimum number of subjects in each category to provide meaningful information to be gathered with that from other countries/settings. The sample includes 10 partners (of any gender) for each group of pregnant and non-pregnant women. All these points are now included in the corresponding paragraph of the manuscript.

9. Page 10, line 20 - What is GBV? This appeared without definition. again, in page 11, line 5.

Apologies. This is gender-based violence and now is defined in the manuscript.

10. Page 15, line 50/51 - salute the authors to think beyond heterosexual partnership. Would suggest encompassing a non-gender approach knowing that only biological females can become pregnant

Thank you for the comment.

VERSION 2 – REVIEW

REVIEWER	Hedriana, Herman University of California Davis, Department of Obstetrics and Gynecology
REVIEW RETURNED	04-Apr-2022

GENERAL COMMENTS	It was interesting to review Cecatti et al's study protocol which aims to understand the needs, attitudes and practices related to sexual and reproductive health (SRH) as it relates to pregnancy, pregnancy prevention and abortion during the COVID-19 pandemic. The protocol is written well, and I am satisfied with the modifications and changes made from the native manuscript. As a study focused on a very diverse international target, in all the sensitivities that is mindful in SRH research, equity and inclusivity is paramount. The word “subject” or “participant” is neutral and does not carry the stigma of discrimination. Abstract – succinct and well stated. Strength and limitations are stated well especially the limitation which is important in the interpretation of study outcome.
--

	Introduction – There is high variability is what “pandemic wave state” a country is at presently. Consider deleting the phrase “currently a second wave (and in some cases a third wave) has initiated everywhere” (page 4, line 20). Page 5, line 27 – consider substituting the phrase “especially married or in union women” to “especially those in established relationship” for a more neutral tone. Line 28, page 6, delete “for” in specific objective #3. Overall, the introduction is informative, and the objectives are clear for the purpose of the study. Methods – Line 48, page 6 – defining male or female partner is okay but reads awkward to me. I suggest “(for the subjects’ partners and health workers)” which seems more equitable. This goes the same for line 58/59 – delete “(males or females)”, also in page 12, line 57/58. Line 49 & 51, consider “subjects” for “women” which is more inclusive (in cases when a trans-man is involved). This true to line 26, page 7, “pregnant subjects” instead of “pregnant women”. So much so that in all lines in which “women” is referred as the target of the study, “subject” is the word I prefer. This is scattered throughout the manuscript. This will be an editorial decision for BMJ. Being a WHO study, I feel the study should have equity and inclusivity for a targeted diverse subject population. Page 8, line 11, “will approach all subjects (or participants) following standard...” Overall, the methods and lay out of the data analysis and management are clear and detailed. Ethics and dissemination – I do not think you need to specify gender “she/he” and leave the term “participant” in place representing equity. Otherwise, this portion is well written.
--	---

VERSION 2 – AUTHOR RESPONSE

Prof. Herman Hedriana, University of California Davis, UC Davis Children's Hospital

Comments to the Author:

It was interesting to review Cecatti et al's study protocol which aims to understand the needs, attitudes and practices related to sexual and reproductive health (SRH) as it relates to pregnancy, pregnancy prevention and abortion during the COVID-19 pandemic. The protocol is written well, and I am satisfied with the modifications and changes made from the native manuscript. As a study focused on a very diverse international target, in all the sensitivities that is mindful in SRH research, equity and inclusivity is paramount. The word “subject” or “participant” is neutral and does not carry the stigma of discrimination.

Abstract – succinct and well stated.

Strength and limitations are stated well especially the limitation which is important in the interpretation of study outcome.

Introduction – There is high variability is what “pandemic wave state” a country is at presently. Consider deleting the phrase “currently a second wave (and in some cases a third wave) has initiated everywhere” (page 4, line 20). We agree on this point and have removed the phrase and the sentence.

Page 5, line 27 – consider substituting the phrase “especially married or in union women” to “especially those in established relationship” for a more neutral tone. This suggestion has been taken up.

Line 28, page 6, delete “for” in specific objective #3. Overall, the introduction is informative, and the objectives are clear for the purpose of the study. Thank you for pointing this out and we have edited the objective to make it clearer.

Methods – Line 48, page 6 – defining male or female partner is okay but reads awkward to me. I suggest “(for the subjects’ partners and health workers)” which seems more equitable. This goes the same for line 58/59 – delete “(males or females)”, also in page 12, line 57/58. These changes have been applied.

Line 49 & 51, consider “subjects” for “women” which is more inclusive (in cases when a trans-man is involved). For line 49, we have to keep it to “women” to identify the subgroup of study participants/subjects who will have the interviews at two time points. For line 51, we have changed it to “subjects”.

This true to line 26, page 7, “pregnant subjects” instead of “pregnant women”. So much so that in all lines in which “women” is referred as the target of the study, “subject” is the word I prefer. This is scattered throughout the manuscript. This will be an editorial decision for BMJ. Being a WHO study, I feel the study should have equity and inclusivity for a targeted diverse subject population. We agree with the Reviewer’s point. We have inserted a footnote to address this concern since there are some instances that using “women” will be clearer.

The footnote reads as:

We recognize that cisgender women ,transgender men, nonbinary, gender-fluid and intersex individuals with a female reproductive system and capable of becoming pregnant may require SRH care. To facilitate readability of this protocol, when referring to all gender diverse people who may require SRH services, we used the word “women” most often, although we also use the term “subjects” and “individual”.

Page 8, line 11, “will approach all subjects (or participants) following standard...” Overall, the methods and lay out of the data analysis and management are clear and detailed. We have followed the suggestion of “participants” but have made a slight adjustment by writing “potential participants”.

Ethics and dissemination – I do not think you need to specify gender “she/he” and leave the term “participant” in place representing equity. Otherwise, this portion is well written. Thank you, this suggestion has been applied.

Reviewer: 1

Competing interests of Reviewer: None